# Biomimetic Nanopillar Silicon Surfaces Rupture Fungal Spores

**DOI:** 10.3390/ijms24021298

**Published:** 2023-01-09

**Authors:** Denver P. Linklater, Phuc H. Le, Arturo Aburto-Medina, Russell J. Crawford, Shane Maclaughlin, Saulius Juodkazis, Elena P. Ivanova

**Affiliations:** 1School of Science, STEM College, RMIT University, Melbourne, VIC 3000, Australia; 2ARC Research Hub for Australian Steel Manufacturing, Wollongong, NSW 2505, Australia; 3BlueScope Steel Ltd., Port Kembla, Wollongong, NSW 2505, Australia; 4Optical Science Centre, Swinburne University of Technology, Hawthorn, Melbourne, VIC 3122, Australia

**Keywords:** antifungal surface, *Aspergillus brasiliensis*, nanopillar surface, biomimetic surface

## Abstract

The mechano-bactericidal action of nanostructured surfaces is well-documented; however, synthetic nanostructured surfaces have not yet been explored for their antifungal properties toward filamentous fungal species. In this study, we developed a biomimetic nanostructured surface inspired by dragonfly wings. A high-aspect-ratio nanopillar topography was created on silicon (nano-Si) surfaces using inductively coupled plasma reactive ion etching (ICP RIE). To mimic the superhydrophobic nature of insect wings, the nano-Si was further functionalised with trichloro(1H,1H,2H,2H-perfluorooctyl)silane (PFTS). The viability of *Aspergillus brasiliensis* spores, in contact with either hydrophobic or hydrophilic nano-Si surfaces, was determined using a combination of standard microbiological assays, confocal laser scanning microscopy (CLSM), and focused ion beam scanning electron microscopy (FIB-SEM). Results indicated the breakdown of the fungal spore membrane upon contact with the hydrophilic nano-Si surfaces. By contrast, hydrophobised nano-Si surfaces prevented the initial attachment of the fungal conidia. Hydrophilic nano-Si surfaces exhibited both antifungal and fungicidal properties toward attached *A. brasisiensis* spores via a 4-fold reduction of attached spores and approximately 9-fold reduction of viable conidia from initial solution after 24 h compared to their planar Si counterparts. Thus, we reveal, for the first time, the physical rupturing of attaching fungal spores by biomimetic hydrophilic nanostructured surfaces.

## 1. Introduction

Synthetic nanomaterials with high-aspect-ratio nanoprotrusions that resemble the epicuticle of insect wings like dragonfly and damselfly exhibit broad spectrum antibacterial properties [1,2,3,4] The bactericidal activity of nanopillar surfaces is characterised by mechanical lysis of the bacterial cell membrane as the cell contacts the surface [1,2,3,4]. This activity also extends to bacterial endospores [4]. Nevertheless, the interactions between biomimetic nanopillar surfaces and filamentous fungal conidia have not yet been characterised. While bacteria demonstrate significant variation in their size, shape, and composition of the cell wall, filamentous fungal spores exhibit even greater diversity. The size of filamentous fungal spores can range from a few micron to tens-of-micron in diameter [5]. Furthermore, spores are characterised by a resilient protective layer on their cell wall that makes them impervious to multiple environmental stressors such as salts, UV, temperature, antifungal molecules, and even physical disruption [6,7]. Therefore, filamentous fungal spores would require a greater deformational stress to be susceptible to mechanical lysis or rupture.

Fungi are a diverse group of eukaryotic organisms that are widespread within nature. Filamentous fungal species such as *Aspergillus* spp. including *A. fumigatus*, *A. terreus*, *A niger*, *A. brasisiensis* and *A. flavus* have been identified as opportunistic pathogens that pose a significant health risk to immunocompromised individuals [8]. These fungi can cause various diseases, and some have been detected in the water of dental equipment. This environment offers suitable conditions for the emergence of fungal biofilms [9]. Antimicrobial resistance (azole resistance) in *A. fumigatus* has been reported to cause invasive infections with high mortality rates [10,11]. This is particularly concerning as the incidences of an invasive human fungal infection often result in a mortality rate greater than 50% [12]. *Aspergillus* spp. have been implicated in aspergillosis of the oral cavity after tooth extraction or endodontic treatment [13,14,15]. Water flow through the tubes of dental units is known to contain multiple microorganisms [16]. This water is a potential source of microbial contamination and a potential threat to the patients’ and professionals’ health. The American Dental Association has established that the bacterial load in the water of dental units must not exceed 200 CFU/mL, but the limit of fungal load has not been recorded [17]. During endodontic treatment, direct contact with contaminated water may cause various respiratory infections, allergies, and infect wounds on the mucosal membranes [18]. Oral aspergillosis is rare, and mandibular aspergillus osteomyelitis is even rarer; however, the outcomes are severe and often involve invasive, multiple surgical debridement procedures and resolution with dental implants [19]. Furthermore, *Aspergillus* spp. are known to grow and proliferate on various bone substitutes [20], implicating the susceptibility of dental biomaterials to contamination by environmental filamentous fungi that are also opportunistic pathogens. In addition, the presence of zinc oxide endodontic sealers into the maxillary sinus are known to increase the risk of filamentous fungal infection in immunocompromised individuals [21]. Thus, fungal contamination needs to be controlled on the surface of dental biomaterials for the prevention of invasive fungal infections in both immunocompromised and immunocompetent individuals.

Early research toward the development of antifungal coatings that could resist filamentous fungi colonisation of surfaces involved the creation of nano-composite coatings consisting of embedded nanosized metal oxide particles such as zinc oxide and copper oxide [22,23,24]. Some other approaches include the surface immobilisation of biocides or functionalised nanoparticles [25,26,27,28,29], the addition of photocatalytic materials, and the physical-chemical modification of the surface [30,31] which we recently reviewed [32].

Surface nanotexturing may present a unique opportunity to develop surfaces that exhibit robust and long-lasting antifungal activity. We recently showed that black damselfly *Calopteryx haemorrhoidalis* wings can prevent the attachment of *A. brasiliensis* spores [33]. Like other Odonates (i.e., damselflies and dragonflies), *C. haemorrhoidalis* wings possess a dense layer of crystalline wax over the surface that creates a characteristic dense nanopillar topography. The combination of wax and surface nanoprotrusions facilitate the entrapment of a layer of air that effectively repels attaching conidia [34,35,36].

In this work, we studied the cell-surface interactions between filamentous fungi *A. brasiliensis* spores and nanotextured silicon surfaces. Herein, the mechanisms of filamentous fungi spore adhesion to nanostructured surfaces were studied by investigating the role of surface wettability on the resulting degree of attachment using a combination of electron and fluorescent microscopy techniques. The cell-surface interface was characterised using FIB milling to allow visualisation of the nanopillar-fungal spore bio-interface.

## 2. Results

### 2.1. Surface Characterisation of Nano-Si Surfaces

The nanopattern present on damselfly wings was replicated on silicon surfaces using reactive ion etching method to create a substrate that contained dense high aspect ratio nanoprotrusions, herein referred to as nano-Si (Figure 1A) [4]. Nano-Si surfaces were further modified via silanisation to produce hydrophobic surfaces (nSi-H). Hydrophilic planar silicon (Si) and silanised planer silicon (Si-H) surfaces were used as controls.

The nanotopography of damselfly and dragonfly wings has been widely recognised to possess broad-spectrum antimicrobial activity in that the nanopillars can not only rupture and kill attaching bacteria, but also have the ability to repel fungal conidia, preventing their attachment [2,4,33,37,38]. The efficacy of the mechano-bactericidal action of the wing nanotopography is governed by a few factors; the nanopillar pattern, i.e., height, spacing (density) and flexural rigidity of the nanopillars, and the cell surface characteristics of the attaching microbe [2,3,39]. The nanopillar height must be such that the adsorbing cell wall cannot reach the base of the nanoprotrusion before the elastic limit of the cell membrane is reached [2]. Thus, for microbes in the order of 5 µm diameter, such as filamentous fungi conidia, taller pillars were hypothesised to have the capacity to induce cell lysis. The height of the nanopillars was controlled according to the period of the etching time, with a greater etching time resulting in the development of taller pillars [37,40]. SEM imaging was used to characterise the surface nanoarchitecture, including height, pillar-to-pillar spacing, and pillar density. Plasma etching of silicon surfaces for 45 min resulted in fabrication of nanostructured surfaces that possessed high aspect ratio nanopillars of approximately 800 nm in height with interpillar spacing of approximately 200 nm, and a pillar density of approximately 5.5 µm^−2^, as determined from SEM imaging (Figure 1B). The typical random distribution and pyramidal shape of nano-Si nanopillars is shown in the SEM and AFM micrographs presented in Figure 1. The as-fabricated nano-Si surfaces were highly wettable, exhibiting a water contact angle (WCA) of ~10° (Figure 1D). All surfaces had a uniform black appearance due to the tapered Si pillars that render a gradual refractive index change [41], hence, achieving anti-reflectivity. To simulate the superhydrophobic properties of insect wings, the nano-Si surfaces were coated with an organofunctional fluorosilane [37,42]. A hydrophobic self-assembled monolayer of silane was achieved, as observed by a water contact angle (WCA) of ~160° and confirmed by XPS (Figure 1D). No change in surface topography was identified, as confirmed through AFM characterisation (Figure 1C and Appendix A). Further AFM analysis and surface roughness parameters of newly fabricated surfaces are included in Table 1. AFM surface roughness values report lower S_max_ values that what is estimated from SEM images due to the highly dense nature of the pattern which restricts the ability to scan the entirety of the high aspect ratio pillar.

### 2.2. Interfacial Interactions of A. brasiliensis Spores with Hydrophilic and Hydrophobic Nano-Si Surfaces

The spore-substratum interface between *A. brasiliensis* spores and the nanopillar substratum were investigated via FIB milling and SEM (Figure 2A). Analysis of tilted SEM micrographs of spores on nSi-H (hydrophobic) surfaces revealed that the spores retained their rounded morphology, and in most cases, the spore body was observed to be ‘hovering’ above the surface. Cross-sections of the spore-nanopillar interface revealed that, indeed, the spore was not in contact with the surface. Further examination revealed that the spore coat and inner membrane were well-preserved. By contrast, the spore morphology after attachment to the nSi (hydrophilic) surfaces appears compromised (Figure 2A), as seen in the tilted SEM images. The FIB cross-sections show the nSi nanopillars were in direct contact with the spore (indicated by the yellow arrows). Sequential milling of the spore-nanopillar interface showed there was obvious damage sustained by the spore coat, together with a loss of membrane integrity evidenced by the lack of defined continuous inner and outer membranes and the incidence of airspace, or holes, between the cell wall and the inner membrane that might point to a loss of turgor pressure and the leakage of the cytosolic content (Figure 3). Multiple points of nanopillar contact, and insertion, are obvious in the SEM micrographs. Thus, we assume that the spore has been lysed by increased contact between the nanopillar surface and spore coat. The super-wetting state of the nano-Si surface would facilitate enhanced contact between the spore coat and nanopillar surface [43]. Micrographs of consecutive milling through the spore-substratum interface have been included in Appendix A.

To confirm whether the interaction between the *A. brasiliensis* spores and hydrophilic nSi surface led to membrane rupture, monitoring of propidium iodide (PI) uptake by fluorescence microscopy was used to confirm cell membrane permeabilisation (Figure 2B,D). PI can only be taken into cells that have become permeable through physical damage, or otherwise [44]. Spores were also counterstained with calcofluor white (spore coat) and Syto 9^®^ (nucleic acids). PI uptake was observed in approximately 55% of those non-germinated (non-germinated spore size confirmed to be 5 µm, Appendix A) spores in contact with the hydrophilic nanostructured surfaces (Figure 2C). By contrast, those spores attached onto the planar Si control surfaces did not exhibit PI uptake (Figure 2C,D and Appendix A). As PI is a non-membrane permeable nucleic acid stain [45], evidence of PI uptake via red fluoresence observed under confocal microscopy indicated that the nano-Si nanopillars were capable of rupturing both the spore coating and the inner plasma membrane of the spores, as observed in the FIB-SEM micrographs.

### 2.3. Assessment of the Structural Integrity, Morphology and Viability of the Fungal Spores during Interactions with Nano-Si Surfaces

To study changes to the structural integrity of spores when they encounter the nano-Si surfaces, an in-depth analysis of spore morphology was carried out using a combination of electron and fluorescence microscopy (Figure 4). Analysis of the resulting SEM images revealed that, over a 24 h incubation period under immersed conditions, the *A. brasiliensis* spores maintained their typical spherical morphology on nSi-H, Si-H and Si surfaces similar to that of spores as collected from the *A. brasiliensis* conidiophores after growth on potato dextrose agar (PDA) plates (Figure 4D and Appendix A). However, on hydrophilic nSi surfaces, an altered spore morphology was clearly evident in the SEM micrographs (Figure 4D, false coloured red). The spores appear lysed, having lost their structural integrity. Similarly, CLSM imaging highlighted that the uniformity of the coat for spores attached to nSi surfaces was also disrupted (Figure 4D, red arrow). Analysis of the morphology of *A. brasiliensis* spores on nano-Si and control surfaces after a 3-day incubation period also highlights that those spores attached on nSi surfaces appear to have deteriorated structural integrity (yellow arrows).

Fluorescence microscopy was used to directly analyse the attachment of *A. brasiliensis* spores to the nano-Si substratum (Appendix A). The number of spores attached to the nano-Si surface was found to be approximately 140.2 ± 73.1 mm^−2^. These values were approximately 5× lower than the number of spores attached to the planar control Si surface (713.3 ± 267.0 mm^−2^) after the same 24 h incubation period (** *p* < 0.01) (Figure 4A). The same trend in attachment patterns were observed after 3 days of incubation, indicating that the fungal spores had a reduced ability to settle and germinate on the nano-Si surfaces (Appendix A).

The metabolic activity of attached spores was also determined using an Alamarblue^™^ cell viability assay, which measures the reduction of resazurin to resofurin by the cell (Figure 4B). Any decrease in the reduction of resazurin may indicate a disturbance in the cellular metabolism [46]. Specifically, the data demonstrated that the spore metabolic activity of conidia attached to the nSi surface was markedly reduced after a 3-day incubation period. These results are consistent with our investigation of the spore-nanopillar interface using FIB-SEM that *A. brasiliensis* spores may be ruptured on nSi but not on nSi-H surfaces. Thus, the nSi surfaces may prevent the spore germination and proliferation through physical rupturing of attached spores, whereas on the planar control Si surfaces, the *A. brasiliensis* spores remained metabolically active and showed a marked increase in their respiratory activity.

A direct plate counting technique was used to assess the number of *viable* spores [47] retrieved from the nano-Si surfaces, and this was compared to that obtained from spores retrieved from the planar Si surface (Figure 4C). The hydrophilic nSi surface exhibited a reduction in the number of viable conidia (4.0 × 10^3^ CFU mL^−1^), in comparison to Si surfaces (3.5 × 10^4^ CFU mL^−1^), corresponding to a ~9-fold reduction of viable conidia after 24 h interaction with the nanopillar surfaces. By contrast, there were only 1.0 × 10^2^ CFU mL^−1^ spores retrieved from the nSi-H surfaces, which is ~35× less than retrieved from Si-H surfaces. By day 3, the number of viable spores retrieved from both nanostructured surfaces were similar to their planar counterparts. These data corroborate the (initial) lesser attachment of *A. brasiliensis* spores to hydrophobised surfaces, as directly confirmed by fluorescence microscopy.

## 3. Discussion

We recently demonstrated that the antifungal properties of *Calopteryx haemorrhoidalis* damselfly wings was driven by the remarkable antiwetting properties exhibited by the nanopillar array present on the wing epicuticle [33]. Sustainable air entrapment on the nano-, micro-, and macro-scales facilitated spore-repellent behaviour that enabled the wings to remain contamination free when immersed in a suspension of *A. brasiliensis* fungal spores. To develop a synthetic substratum nanotopography that would show fungicidal propensity toward large micro-organisms, such as filamentous fungi spores that possess a rigid, multilayered spore coat, we used etched silicon substrates possessing nanopillars ~800 nm in height. The anti-wetting properties of insect wing surfaces were then mimicked by surface functionalisation with a self-assembled monolayer of fluoro-silane molecules [37]. Analysis of non-germinated *A. brasiliensis* spore interactions with both superhydrophobic and superhydrophilic nanostructured surfaces was achieved using a combination of fluorescence and electron microscopy, and classical microbiological techniques.

In order to form a biofilm, a microorganisms must first undergo attachment to a substrate [48]. In case of filamentous fungi, initiation of the biofilm starts with the germination of spores attached to a substrate surface. Recently the antifungal capabilities of nanostructured surfaces toward *Candida albicans* and other yeasts were shown to result from a purported physical mechanism, sometimes in addition to the toxic activity of generated strong reactive oxygen species (ROS) [49,50,51,52,53]. For example, Xie et al. reported that *C. albicans* cells were mechanically ruptured when interacting with ZnO nanostructured substrates (a polygonal column structure with a height of 3–5 μm and a width of 100–200 nm) [54]. Upon interaction with ZnO-TiO_2_-nanostructured surfaces, it was discovered that the *A. flavus* conidia lost their typical round morphology [55]. In addition to the intrinsic mechanism of the ROS-mediated antimicrobial effects, the physical orientation (an extended tubular shape) of MoS_2_ has also been shown to play a substantial role in imparting the fungicidal effects of these surfaces toward *Alternaria alternata* cells [56]. In another example, the spores of the filamentous fungi *A. fumigatus* attached onto the nanopillar topography of poly(methyl methacrylate) (PMMA) substrates were observed to have lost their structural integrity, appearing either half or fully deflated [57]. Nevertheless, to the best of our knowledge, the *fungicidal action* of nanostructured hydrophilic surfaces toward filamentous fungi spores has never been previously demonstrated.

In this work, exposing *A. brasiliensis* fungal spores to hydrophilic nano-Si surfaces achieved substantial reductions in spore viability as determined by assessment of cell metabolic activity and plate counting technique (Figure 4B,C). Importantly, *A. brasiliensis* conidia that were attached to nSi surfaces were found to be metabolically inactive. We hypothesise that the hydrophilicity of the nano-Si surfaces led to an increased contact between the spores and the nanopillar surface, resulting in the rupture of the spore coat and inner membrane, leading to cell lysis and death (Figure 1). The biointerface between fungal conidia and surface nanopillars was evaluated using FIB-SEM in order to obtain an insight into the fungicidal properties of the hydrophilic nanostructured surfaces. We verified that a direct interaction took place between the spores and the nanopillars present on the nano-Si surface. The spore coat was adhered to the tips of the nanopillars and the spore coat and membrane was ruptured at the points between nanopillar adhesion. Top-view SEM morphological analysis of the spores adhering to the nano-Si revealed that the spores appeared disturbed, with some displaying a flattened and stretched morphology (Figure 2A, second panel and Figure 4D, false coloured red). The spore coat is composed of glucans, glycoproteins, and chitin, with the rigidity of the cell wall resulting from the unique composition of the β-1,3-glucan and chitin [58,59]. Both the outer spore coat, and the inner membrane appeared to be compromised when in contact with the nano-Si substrate. The loss of membrane integrity was corroborated by the uptake of PI by *A. brasiliensis* spores attached to nSi surfaces (Figure 2B–D). Therefore, we hypothesise that the physical contact between the Si nanopillars contributed to the lysis and death of *A. brasiliensis* conidia attached to these surfaces (Figure 2 and Figure 4D). These findings demonstrate that surfaces containing high aspect ratio nanopillars can play a significant role in inhibiting the degree of fungal attachment and also be fungicidal to the attaching cells through a physical rupture mechanism, similar to what has been observed for bacteria attached on high aspect ratio nanostructured surfaces [4].

Contrastingly, the *antibiofouling* characteristics of superhydrophobic nSi-H surfaces were demonstrated by a reduction in the attachment of fungal spores; notably, the spore density was found to be 5× lower than observed on planar Si surfaces (Figure 4A). The spores displayed a reduced degree of interaction with the nanopillar surface, as evidenced by the presence of an air layer underneath the cell body (Figure 2A).

The loose association of spores to the nSi-H surface may facilitate a self-cleaning effect as *A. brasiliensis* spores may be easily displaced by an external force which is consistent with the interfacial behaviour of *A. brasiliensis* spores observed by Ivanova et al. on damselfly wings [33]. By contrast, the attachment propensity of spores on hydrophilic nSi surfaces was found to be increased in comparison to the hydrophobised nanostructured Si but was still lower than that observed for their respective planar Si counterparts. Our data agrees with recent research that found that biomimetic chitosan hydrogel nanopillars, with an average aspect ratio of 2.1, inhibited the attachment of the filamentous fungi *Fusarium oxysporum* by 99% compared to that observed on a non-structured surface after a 24 h incubation period [48]. Cell-surface interactions represent part of a very complex attachment process, which is governed both by surface physicochemical characteristics and by the cell surface characteristics, i.e., hydrophobicity and surface charge. For example, more hydrophobic cell walls may be more greatly attracted to hydrophobic surfaces [60]. However, as per our previous experiments [33], *A. brasiliensis* spores have a zeta potential of −10.7 ± 1.5 mV and a hydrophobicity index (HI) of 0.71 ± 0.07. WCA measurements of fungal conidia were found to be 107.2 ± 5.4°. These data suggest that the spore coat of *A. brasiliensis* conidia is only moderately hydrophobic and, indeed, may not display an enhanced affinity for hydrophobic surfaces.

## 4. Materials and Methods

### 4.1. Fabrication of Nano-Si Surfaces

P-type boron doped silicon wafers were rinsed in isopropanol and dried with nitrogen gas before being subjected to mask-less plasma etching using a Samco RIE inductively coupled plasma etching system (Model: RIE-101iPH, Samco Inc., Fushimi-ku, Kyoto, Japan). Gases SF_6_ and O_2_ were set at flow rates of 35 and 45 sccm, respectively. ICP RIE power was 150 W and bias power was 15 W. Etch time was 45 min. Following etching, the surfaces were cleaned with 10% H_2_SO_4_ sonication for 10 min.

Nano-Si surfaces were hydrophobised by first exposing to 1 min of O_2_ plasma to generate hydroxyl (Si−OH) surface groups. Then, the substrata were placed in a glass Petri dish inside a desiccator at room temperature, and 100 μL of trichloro(1H,1H,2H,2H-perfluorooctyl)silane was placed beside the samples. The desiccator was placed under vacuum for 1 h. After venting, the samples were rinsed twice with chloroform, twice with ethanol and then dried with gentle nitrogen gas flow.

### 4.2. Atomic Force Microscopy (AFM)

The surface topography of the substrata was characterised using a NanoWizard^®^4 tip scanning AFM (JPK BioAFM Business, Bruker NanoGmbH, Berlin, Germany). The AFM head was placed on an upright optical microscope (IX81, Olympus, Japan), and the tests were conducted in an acoustic hood and on an active vibration isolation table (Accurion, Goettingen, Germany). The scans were conducted in an air-conditioned environment at a temperature of around 22 °C using an n-type antimony doped silicon probe (SICON, AppNano, Mountain View, CA, USA) in the Quantitative Imaging™ (Qi) mode of JPK. The cantilever’s spring constant ranged between 0.1 and 0.6 N m^−1^. Triplicate scans of each surface were taken across scan regions of 10 × 10 µm^2^, and a set of roughness parameters, including average roughness (S_a_), root mean square roughness (S_q_), and maximum height, were then calculated (S_max_). Gwyddion (ver. 2.53) was used to analyse the AFM images, and Amira Avizo software (Thermo Fisher Scientific, Waltham, MA, USA) was utilised to create the three-dimensional scans.

### 4.3. Surface Wettability

The static water contact angle was determined for each substrate using the sessile drop technique. Contact angles were determined using a Phoenix-MT(T) instrument (SEO Co., Yongin, Gyeonggi, Republic of Korea) coupled with SurfaceWare 9 software. Static water contact angles were measured within two seconds of the 10 μL droplet. The results are an average of five independent measurements taken on each substratum.

### 4.4. X-ray Photoelectron Spectrometry

The elemental analysis of the sample surfaces was conducted using a Thermo Scientific K-alpha X-ray photoelectron spectrometer (XPS) (Thermo Fischer Scientific, USA). The K-alpha EPS instrument was equipped with a monochromatic X-ray source (Al Kα, hν = 1486.6 eV) operating at 150 W. Photoelectrons emitted at 90° to the surface from an area of 400 × 400 μm^2^ were analysed at 200 eV for survey spectra and then at 50 eV for region spectra. Survey spectra were recorded at 1.0 eV per step, while the region spectra were recorded at 0.1 eV per step. The relative atomic concentrations of elements obtained by XPS were measured using the peak area in the specified high-resolution region and the instrument-specific sensitivity parameters. High-resolution scans were performed across each of the titanium 2p, carbon 1s and oxygen 1s peaks.

### 4.5. Microorganism, Growth Conditions and Experimental Set-Up

*Aspergillus brasiliensis* ATTC 9642^™^ was acquired from the American Type Culture Collection (ATCC, Manassas, Virginia, U.S.A.). The fungal stocks were prepared in 20% glycerol nutritional broth (Oxoid - Thermo Fisher Scientific, Waltham, MA, USA) and kept at −80 °C. Fungi from the glycerol stock were growth on potato dextrose agar (PDA) plates (Neogen^®^ Culture Media, Bundamba, QLD, Australia). These plates were incubated at 27 °C for a week until the fungal spores were fully growth on top of the agar plates. Sterilised MilliQ water (resistivity: 18.2 MΩ cm, 25 °C) was pour onto the fungal agar plates and then the plates were shaken for removing spores from the top layer of the agar plate. To obtain conidial suspension, this suspension was filtered using Whatman^®^ Grade 1 filter paper (Sigma-Aldrich Pty Ltd., Melbourne, VIC, Australia). The final density of the working conidial suspension was adjusted to 1 × 10^5^ mL^−1^ using the TC20 Automated Cell Counter (Bio-Rad, South Granville, NSW, Australia).

Hydrophobic nano-Si (nSi-H), hydrophobic Si (Si-H), hydrophilic nano-Si (nSi) and hydrophilic Si (Si) were cut into 0.5 cm^2^ pieces using a diamond pen (ProSciTech Pty Ltd., Kirwan, QLD, Australia). Surfaces were sterilised by sequential sonication in 100% and 70% ethanol (EtOH) (Chem-supply, Gillman, South Australia, Australia) for 15 min each. The studied surfaces were gently dried with nitrogen gas flow and placed in a desiccator to prevent moisture adsorption until required.

The surfaces were immersed in a suspension of 1 × 10^5^ of *A. brasiliensis* conidia and incubated at 25° for 24 h. Afterward, samples were collected and analysed as outlined below.

### 4.6. Confocal Laser Scanning Microscopy

A Zeiss LSM 880 Airyscan upright CLSM (Carl Zeiss Microscopy, Oberkochen, Germany) operated with a 63× water-immersion objective (ZEISS 60x/1.0 VIS-IR) was used for visualisation of *A. brasiliensis* attachment on surfaces after 1-Day and 3-Day incubation period. The examined surfaces were carefully washed 3× with MilliQ water (2 mL each time) to remove non-attached cells and then put in a 3.5 mm glass-bottomed Petri dish filled with 3 mL MilliQ water. In terms of fungal attachment study, Calcofluor white (Biotium, Fremont, CA, USA) (blue colour) was used to stain the fungal spores, which binds to cellulose and chitin in fungal cell wall, and NucSpot (Biotium, Fremont, CA, USA) (green colour) was used to stain the fungal nuclei containing DNA. The adhesion of *A. brasiliensis* conidia to surfaces was quantified using 15 distinct fields of view of 135 × 135 µm^2^ for each sample. The ImageJ 1.52a Cell Counter plugin was used to recognise and count cells on CLSM micrographs. Results were derived from the average of at least 3 independent experiments, containing two replicates for each experiment.

For assessment of cell viability, LiveDead staining was carried out using a combination of Syto9 (Thermo Fischer Scientific, USA), a membrane permeable nucleic acid stain (green colour) and Propidium iodide (PI) (Thermo Fischer Scientific, USA), a non-membrane permeable nucleic acid stain (red colour). Cell membrane damage was determined by the uptake of PI.

Regarding CLSM settings, ZEN Black software (Carl Zeiss Microscopy, Germany) was used for CLSM imaging and operation. The information of the dyes used in this investigation were selected from the Smart Setup database integrated in the software. In addition, the Z-stack feature was enabled to capture the whole attached fungus spores and adjustments were made to the laser’s intensity until neither blur nor noise were visible. Scanning speed was maintained at 7 with the pixel dwell at 1.58 μs and the scan time at 7.75 s.

### 4.7. Scanning Electron Microscopy (SEM)

The samples with attached *A. brasiliensis* spores were fixed with 2.5% glutaraldehyde for 45 min and then washed 3× with 0.1 M cacodylate buffer containing 2 mM calcium chloride. The surfaces were plunge-frozen in liquid nitrogen and then freeze-dried for SEM imaging. Prior to imaging, 5 nm of iridium (Ir) was sputtered on the surface. Imaging was performed using an FEI Nova NanoSEM at 5kV.

### 4.8. Focused Ion Beam SEM

FIB milling of *A. brasiliensis* spores incubated on the nano-Si surfaces was performed using an FEI Scios dual-beam FIB system. The samples were fixed with 2.5% glutaraldehyde for 45 min and then washed 3× with 0.1 M cacodylate buffer containing 2 mM calcium chloride. The samples were post-fixed with 2% osmium tetroxide and 1.5% potassium ferrocyanide for 1 h at room temperature and then washed extensively. Following this, samples were incubated for 20 min with thiocarbohydrazide, used as a mordant. A second staining was completed using 2% osmium tetroxide for 30 min. Afterwards, the samples underwent ethanol dehydration over ice and drying using a critical point drying system, the sample surfaces were finally coated with a Pt protection layer using an e-beam Pt deposition process prior to milling.

### 4.9. Statistical Analysis

The data were examined to validate their normal distribution and homogeneity of variance using the Shapiro–Wilk and Levene’s tests, respectively, using SPSS Statistics 26 software (IBM, New York, NY, USA). Values was given as the mean value ± one standard deviation. Statistical data were evaluated using one-way ANOVA. Differences between the mean values were evaluated using Tukey’s range test. Values were deemed statistically significant differences if p-values were less than 0.05 (* *p* < 0.05, ** *p* < 0.01, and *** *p* < 0.001).

## 5. Conclusions

In this study, the influence of hydrophilicity on the antifungal properties of nanostructured surfaces was revealed for the first time. Both superhydrophobic and superhydrophilic nanostructured Si surfaces displayed antifungal behaviour; however, the behaviour observed was distinct depending on the wettability of the substrate. The combination of superhydrophobicity and high-aspect ratio nanopillar pattern on the surface of hydrophobised nanostructured Si was shown to play an essential role in determining its antifouling capabilities, as it significantly reduced the extent of fungal adhesion in an aqueous environment, leading to a marked reduction in spore association and/or attachment over a several-day incubation period in comparison with control planar surfaces. By contrast, the results obtained in this study revealed that the viability of *A. brasiliensis* spores were directly affected by direct interaction with the hydrophilic nanopillar surface, whereby the nanopillars were able to rupture the cell wall of conidia, leading to the uptake of propidium iodide into the cells and reduced metabolic activity. This work demonstrates that nanopillar surface hydrophilicity greatly contributes to the fungicidal activity of nano-Si surfaces. By a combination of the superhydrophobicity and nanopillar surface topography, we have duplicated the antibiofouling effect of damselfly wings toward *A. brasiliensis* conidia. Conversely, we have also determined that hydrophilic nanopillar-surfaces may achieve the physical rupture and cell death of *A. brasiliensis* conidia attaching to these surfaces. This study enhances the present understanding of the effects of surface nanotopography and surface wettability on fungal spore attachment and proposes a potential surface that has the potential to be applied to the development of antifungal biomaterials, including dental biomaterials. Future work is necessary to apply the surface defined in the current study to relevant biomaterials.

## Data Availability

Data could be provided upon request.

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
