# Peer review of "Biomimetic Nanopillar Silicon Surfaces Rupture Fungal Spores"

_ijms, 2023, doi:10.3390/ijms24021298_

Round 1

Reviewer 1 Report

In the research article „Biomimetic nanopillar silicon surfaces rupture fungal spores” by Linklater et al. the authors describe the effect of high aspect-ratio nanopillar surfaces of different wettability on fungal spore attachment, membrane integrity and metabolic activity. The effect is already known from several bacteria, but it is interesting to see the extent to fungal spores. In general, the experiments are nicely conducted and the conclusion is supported by the results. Nevertheless, their overall presentation and the connection to the topic of this special issue need to be improved.

1) The article has been submitted to the special issue Dental Biomaterial, but within the manuscript, there is only one sentence having a connection to dentistry. I would therefore like a stronger focus in the introduction on why this topic is relevant for dentistry and especially biomaterials. Also, it should be explained, why A. brasiliensis was chosen as, to my knowledge, this is no dental fungi. The same applies for the conclusion. What do these results mean for dental biomaterials? Can the surfaces be directly applied? What improvements might have been to considered? What would be the next steps to make the material a dental biomaterial?

2) The results and figures are sometimes confusing. In section 2.1, several figure cross references are wrong. In section 3.3, figures are not mentioned consecutive and there is a lot of jumping back and forth with the supplementary material (e.g., Figure 4C, Figure S5-9, Figure 4A). I would highly suggest to combine at least Figure 4 with Figure S4, S5 and S9. The results are described together and the reader needs the comparisons to follow. An then, I would try to arrange the panels in the order mentioned in the text. Or change the text otherwise. For figures that remain in the supplementary, I would also like to get the main result in the text, not only, that the figure can be found somewhere.

3) Regarding the results, one of the main findings seems to be the wrinkled surface on spores on nSi. But to me, the same wrinkles can be seen on all surfaces of Figure S5. What is the explanation? Also, the experiment for PI uptake is a nice explanation of the mechanism. However, there is no quantification, only one cell per condition is shown. Either, this finding should be supported with more data, or the relevance of this observation should be reduced. Why is the result for Figure S10 given in the discussion and not in the results?

5) In the results, the red line of arguments could be clearer. From line 326 to 357, first the hydrophobic surfaces are discussed, then the hydrophilic, and then again the hydrophobic ones.

4) For statistics, sometimes, bars look strongly different, but don’t have signs of significance (e.g., Figure 4B). How is this explained? Also, the number of repetitions should be given for each experiment in the Materials and Methods section.

5) There are concluding sentences at early sections of the results (e.g. 2.2) that are not supported with data at this stage of the manuscript. I would move them to the discussion. The same applies to Figure 2B, which is not at all mentioned in the text. This figure seems to be the final conclusion to the manuscript. I would place it next to the conclusion. Here, it would be nice to have different colors for nSi-H and nSi.

Some minor points:

1) Labelling in Figure 1A and B is hard to read.

2) The abbreviations used (Si, nSi, Si-H, nSi-H) should be introduced at the beginning of the results and then used consistently in the text.

3) In section 2.2, spores are mentioned for the first time. Please also give the fungi strain here.

4) There are a lot of information in the sentence from line 156-160. I would recommend splitting.

5) The conclusion of section 2.2 in the last sentence belongs into the discussion.

6) Figure 3 could have less arrows. The cover the really interesting part of the figure.

7) Microscopy settings for CLSM should be given.

Author Response

In the research article „Biomimetic nanopillar silicon surfaces rupture fungal spores” by Linklater et al. the authors describe the effect of high aspect-ratio nanopillar surfaces of different wettability on fungal spore attachment, membrane integrity and metabolic activity. The effect is already known from several bacteria, but it is interesting to see the extent to fungal spores. In general, the experiments are nicely conducted and the conclusion is supported by the results. Nevertheless, their overall presentation and the connection to the topic of this special issue need to be improved.

  • The article has been submitted to the special issue Dental Biomaterial, but within the manuscript, there is only one sentence having a connection to dentistry. I would therefore like a stronger focus in the introduction on why this topic is relevant for dentistry and especially biomaterials. Also, it should be explained, why A. brasiliensis was chosen as, to my knowledge, this is no dental fungi. The same applies for the conclusion. What do these results mean for dental biomaterials? Can the surfaces be directly applied? What improvements might have been to considered? What would be the next steps to make the material a dental biomaterial?

As suggested by the reviewer, we have applied a stronger focus in the introduction to filamentous fungi infections of the oral cavity and implication for dental biomaterials.

Aspergillus spp. have been implicated in aspergillosis of the oral cavity after tooth extraction or endodontic treatment (13-15). Water flowthrough the tubes of dental units is known to contain multiple microorganisms (16). This water is a potential source of microbial contamination and a potential threat to the patients' and professionals' health. The American Dental Association has established that the bacterial load in the water of dental units must not exceed 200 CFU/mL, but the limit of fungal load has not been recorded (17). During endodontic treatment, direct contact with contaminated water may cause various respiratory infections, allergies, and infect wounds on the mucosal membranes (18). Oral aspergillosis is rare, and mandibular aspergillus osteomyelitis is even rarer; however the outcomes are severe and often involve invasive, multiple surgical debridement procedures and resolution with dental implants (19). Furthermore, Aspergillus spp. Are known to grow and proliferate on various bone substitutes(20), implicating the susceptibility of dental biomaterials to contamination by environmental filamentous fungi that are also opportunistic pathogens. Thus, fungal contamination needs to be controlled on the surface of dental biomaterials for the prevention of invasive fungal infections in both immunocompromised and immunocompetent individuals.

  • The results and figures are sometimes confusing. In section 2.1, several figure cross references are wrong. In section 3.3, figures are not mentioned consecutive and there is a lot of jumping back and forth with the supplementary material (e.g., Figure 4C, Figure S5-9, Figure 4A). I would highly suggest to combine at least Figure 4 with Figure S4, S5 and S9. The results are described together and the reader needs the comparisons to follow. An then, I would try to arrange the panels in the order mentioned in the text. Or change the text otherwise. For figures that remain in the supplementary, I would also like to get the main result in the text, not only, that the figure can be found somewhere.

As suggested by the referee, we have combined Figure 4 with Fig S4, S5 and S9. The text has been changed accordingly to follow the flow of the figure.

  • Regarding the results, one of the main findings seems to be the wrinkled surface on spores on nSi. But to me, the same wrinkles can be seen on all surfaces of Figure S5. What is the explanation? Also, the experiment for PI uptake is a nice explanation of the mechanism. However, there is no quantification, only one cell per condition is shown. Either, this finding should be supported with more data, or the relevance of this observation should be reduced. Why is the result for Figure S10 given in the discussion and not in the results?

We do not only observe wrinkling on the surface of the spores, but they seem to be flattened on the nanostructure surfaces. CLSM micrographs of spore morphology on day 1 (left panels) and day 3 (right panels) both show degradation of the normal spore morphology. Spores incubated on non-textured surface types show highly rounded spore morphology under wet (CLSM) conditions. Somewhat wrinkling surface of some spores in SEM images may be attributed to drying effects of some damaged spores during the preparation for imaging. However, these artefacts are not present in fluorescence images of intact spores as seen in Figure 4.

5) In the results, the red line of arguments could be clearer. From line 326 to 357, first the hydrophobic surfaces are discussed, then the hydrophilic, and then again the hydrophobic ones.

4) For statistics, sometimes, bars look strongly different, but don’t have signs of significance (e.g., Figure 4B). How is this explained? Also, the number of repetitions should be given for each experiment in the Materials and Methods section.

Statistics have been performed on all the data presented in the graph using a one-way ANOVA and post-hoc Tukey’s multiple comparisons test. All significance is now shown in each graph. The materials and methods section has been updated accordingly.

5) There are concluding sentences at early sections of the results (e.g. 2.2) that are not supported with data at this stage of the manuscript. I would move them to the discussion. The same applies to Figure 2B, which is not at all mentioned in the text. This figure seems to be the final conclusion to the manuscript. I would place it next to the conclusion. Here, it would be nice to have different colors for nSi-H and nSi.

Some minor points:

  • Labelling in Figure 1A and B is hard to read.

The labelling has been removed to the caption.

2) The abbreviations used (Si, nSi, Si-H, nSi-H) should be introduced at the beginning of the results and then used consistently in the text.

3) In section 2.2, spores are mentioned for the first time. Please also give the fungi strain here.

The first sentence has been changed to ‘The spore-substratum interface between A. brasiliensis spores and the nanopillar substratum were investigated via SEM and FIB milling (Figure 2).’

4) There are a lot of information in the sentence from line 156-160. I would recommend splitting.

As suggested, we have split into 3 sentences like so:

Multiple points of nanopillar contact, and insertion, are obvious in the SEM micrographs. Thus, we assume that the spore has been lysed by increased contact between the nanopillar surface and spore coat. The super-wetting state of the nano-Si surface would facilitate enhanced contact between the spore coat and nanopillar surface.

5) The conclusion of section 2.2 in the last sentence belongs into the discussion.

As suggested, we have moved the last sentences to the discussion section.

6) Figure 3 could have less arrows. The cover the really interesting part of the figure.

Some arrows have been removed as suggested and others added that point out the interesting sections of the image.

7) Microscopy settings for CLSM should be given.

We have provided additional details for the CLSM settings.

Reviewer 2 Report

In this manuscript, inspired by insect wings, the authors developed biomimetic nanopillar surface. Obtained materials showed good antifungal property and they have an important potential for several applications. The topic of the paper is of large interest in biomimetic materials and surface interface. The results and analyses are both interesting and valuable. As such, the paper may be of interest to the readers of IJMS. However, the article has a few deficiencies, and it needs some corrections before being considered for publication.

- Durability of thin films against environmental disturbances is very important for real-world applications. How robust is the fluorosilane nanocoating?

- Please, provide not only brand information (Samco RIE instrument) of plasma etching equipment, but also model information.

Author Response

In this manuscript, inspired by insect wings, the authors developed biomimetic nanopillar surface. Obtained materials showed good antifungal property and they have an important potential for several applications. The topic of the paper is of large interest in biomimetic materials and surface interface. The results and analyses are both interesting and valuable. As such, the paper may be of interest to the readers of IJMS. However, the article has a few deficiencies, and it needs some corrections before being considered for publication.

- Durability of thin films against environmental disturbances is very important for real-world applications. How robust is the fluorosilane nanocoating?

Fluorosilane nanocoatings are recognized for their durability and stability. According to our internal testing, fluorosilane-coated surfaces may maintain their hydrophobicity for months at ambient temperature. For sterilising purposes, fluorosilane-coated surfaces were also washed and sonicated with water and ethanol; nevertheless, there was no variation in their hydrophobicity even after 4-5 cycles. Zhang et al. demonstrated that after being subjected to 200 cycles of abrasion and ultrasonication for many hours, the fluorosilane coating can withstand and retain its superhydrophobic qualities [1]. Additionally, it has superior acid and alkali corrosion resistance. Therefore, these properties make its ideal for use on a wide range of medical and industrial products.

Reference:

[1] Zhang, Z., Ge, B., Men, X. and Li, Y., 2016. Mechanically durable, superhydrophobic coatings prepared by dual-layer method for anti-corrosion and self-cleaning. Colloids and Surfaces A: Physicochemical and Engineering Aspects490, pp.182-188.

- Please, provide not only brand information (Samco RIE instrument) of plasma etching equipment, but also model information.

As suggested, we have added the model information for the Samco RIE instrument in materials and methods.

Reviewer 3 Report

In the manuscript “Biomimetic Nanopillar Silicon Surfaces Rupture Fungal Spores”, Linklater et al. demonstrated a bioinspired surface with anti-fungal properties, where a high-aspect-ratio nanopillar topography is able to rupture fungal spore membranes, reducing the organism attachment drastically. The manuscript is well-written and carefully organized. The FIB-SEM micrographs are beautiful. However, a few suggestions can be made:

·         Line 81-85. The mechanism to reduce attachment due to decreased contact force between organism-surface is not well introduced. Suggested papers to enhance the discussion:

o   DOI: 10.1021/acsami.1c22205

o   DOI: 10.1088/1748-3190/ac060f

·    Line 99. Please add a reference to the statement: “…render a gradual refractive index change…”

·       Line 104. Is it surface topology or surface topography?

·     Figure 1. The figure could be more attractive. For instance, the AFM line profiles in Fig 1C and 1D are unnecessary. Not only do they not give out any information, but also it is not indicated where the line profile was extracted in the main figure. The scale bar in the 3D representation does not have values, only high-medium-low. Fig 1B (right) indicates the dimensions, while Fig 1A (right) does not. Adding these values only in the main text and figure caption is suggested.

·     The roughness parameters are a piece of essential information for anti-fungal, anti-fouling, and self-cleaning surfaces. It is suggested to transfer the table from Fig. S1B to the main text.

·         Line 403. Regarding referencing the Samco RIE instrument, it is suggested to use the same style as shown in the following sections: (Model, Brand, Country).

·         Line 495. Why is a second staining with osmium tetroxide necessary?

Author Response

In the manuscript “Biomimetic Nanopillar Silicon Surfaces Rupture Fungal Spores”, Linklater et al. demonstrated a bioinspired surface with anti-fungal properties, where a high-aspect-ratio nanopillar topography is able to rupture fungal spore membranes, reducing the organism attachment drastically. The manuscript is well-written and carefully organized. The FIB-SEM micrographs are beautiful. However, a few suggestions can be made:

  • Line 81-85. The mechanism to reduce attachment due to decreased contact force between organism-surface is not well introduced. Suggested papers to enhance the discussion:
  • DOI: 10.1021/acsami.1c22205
  • DOI: 10.1088/1748-3190/ac060f

Thanks for your suggestion, we have added these references into the text (refs 38-39).

  • Line 99. Please add a reference to the statement: “…render a gradual refractive index change…”

As suggested, we have added references for this sentence.

  • Line 104. Is it surface topology or surface topography?

We have changed it to surface topography.

  • Figure 1. The figure could be more attractive. For instance, the AFM line profiles in Fig 1C and 1D are unnecessary. Not only do they not give out any information, but also it is not indicated where the line profile was extracted in the main figure. The scale bar in the 3D representation does not have values, only high-medium-low. Fig 1B (right) indicates the dimensions, while Fig 1A (right) does not. Adding these values only in the main text and figure caption is suggested.

As suggested by the reviewer we have removed the line profiles from Fig 1. We have added height values to the scale bar for the 3D AFM representations of each surface. The surface dimensions from the SEM image has been moved to the figure caption.

  • The roughness parameters are a piece of essential information for anti-fungal, anti-fouling, and self-cleaning surfaces. It is suggested to transfer the table from Fig. S1B to the main text.

The table for roughness values of each surface has been moved to the main text.

  • Line 403. Regarding referencing the Samco RIE instrument, it is suggested to use the same style as shown in the following sections: (Model, Brand, Country).

The detailed information for Samcon RIE instrument has been added in Materials and methods.

  • Line 495. Why is a second staining with osmium tetroxide necessary?

Typically, aldehydes are used for the first fixation of biological specimens prior to SEM processing. In our research, glutaraldehyde was used to cross-link the proteins in the sample. Especially for FIB-SEM sample preparation, a second fixation step is frequently applied. Osmium tetroxide is the most prevalent secondary fixative, and it has the benefit of retaining lipid membranes, which are not maintained by aldehyde fixation solely. In addition to functioning as a stain, it imparts a substantial degree of contrast and conductivity to the sample.

Round 2

Reviewer 1 Report

The authors have fullfilled all comments from the previous review.